# All-Optical Nanosensor for Displacement Detection in Mechanical Applications

**DOI:** 10.3390/nano12224107

**Published:** 2022-11-21

**Authors:** Lorena Escandell, Carlos Álvarez-Rodríguez, Ángela Barreda, Ramón Zaera, Braulio García-Cámara

**Affiliations:** 1Group of Displays and Photonics Applications, Carlos III University of Madrid, Avda. de la Universidad, 30, Leganés, 28911 Madrid, Spain; 2Institute of Solid State Physics, Friedrich Schiller University Jena, Helmholtzweg 3, 07743 Jena, Germany; 3Department of Continuum Mechanics and Structural Analysis, Carlos III University of Madrid, Avda. de la Universidad, 30, Leganés, 28911 Madrid, Spain

**Keywords:** optical sensor, nanowires, displacement sensor, light scattering resonances

## Abstract

In this paper, we propose the design of an optical system based on two parallel suspended silicon nanowires that support a range of optical resonances that efficiently confine and scatter light in the infrared range as the base of an all-optical displacement sensor. The effects of the variation of the distance between the nanowires are analyzed. The simulation models are designed by COMSOL Multiphysics software, which is based on the finite element method. The diameter of the nanocylinders (*d* = 140 nm) was previously optimized to achieve resonances at the operating wavelengths (*λ =* 1064 nm and 1310 nm). The results pointed out that a detectable change in their resonant behavior and optical interaction was achieved. The proposed design aims to use a simple light source using a commercial diode laser and simplify the readout systems with a high sensitivity of 1.1 × 10^6^ V/m^2^ and 1.14 × 10^6^ V/m^2^ at 1064 nm and 1310 nm, respectively. The results may provide an opportunity to investigate alternative designs of displacement sensors from an all-optical approach and explore their potential use.

## 1. Introduction

Current requirements in sensing technology are demanding further research about their design and development to increase the sensitivity and improve the on-chip integration. In this sense, all-optical approaches and applications of silicon nanowires are providing in recent decades promising solutions in topics such as nanoelectronics, optoelectronics, and sensing/detection [1,2,3].

Micro- and more recently, nano-optoelectromechanical systems (M/NOEMS) have been widely used to design a huge number of sensors for a large variety of applications, such as health care, food quality control, or security [4,5,6,7,8,9]. They are mainly focused on sensing displacement, mass [10], or force [11] as a consequence of the target phenomena such as molecular bindings, mechanical vibrations, or external stress. As an example, MOEMS and NOEMS are present in devices such as atomic force microscopes [12], accelerometers [13], or biosensors [14]. Current trends show that optical solutions provide a better performance in terms of sensitivity, resolution, and power consumption compared to other sensing methods (e.g., piezoelectric or capacitive) [15]. In this sense, we can find in the literature many proposed sensors based on interferometers [16], nanocavities [17], photonic crystals [18], or nanocantilevers [19], offering quite interesting performances. However, the readout system’s complexity and limited dynamic range or bandwidth of these proposals make it necessary to continue searching for alternatives [20].

Due to the small signal produced by these nanostructures, resonant behaviors are usually looked for in order to optimize the signal-to-noise ratio. Sensors based on plasmonic resonances or Mie resonances can be found in almost every sensing field, from biosensors for lab-on-chip applications [21,22] to integrated NOEMS [23]. The phenomena appear in scattered light by either metallic or high-refractive-index dielectric nanostructures producing a remarkable increment of the scattered signal. These resonances are highly dependent on the properties of the nanostructures (e.g., refractive index, size, and shape) and the surrounding medium. Additionally, the presence of other resonant nanostructures can produce joint effects depending on the inter-distances.

Those dependencies make resonant nanostructures suitable for sensing applications mainly based on variations of the refractive index of the surrounded medium [24]. That variation may be induced by the presence of target analytes [25] or changes in ambient conditions, such as temperature [26]. These resonant nanoparticles are not usually used in an individual way. On the contrary way, they are used to compose arrays or metasurfaces [27], taking advantage of not only the resonant nanoparticles but also other joint effects (e.g., diffractive modes) [28]. This kind of sensor provides ultra-high sensitivities and ultra-low detection limits with a small footprint and low power consumption making them ideal for the next sensing generation [29].

The research field of plasmomechanics [30] is worth mentioning, where the fundamental coupling between plasmon resonances and mechanical motions is analyzed. Indeed, the excitation of plasmon resonances in metallic nanoparticles can be spectrally shifted under an external mechanical displacement as the sensor described in reference [31] or produce mechanical motions due to the optical forces induced by the strong confinement of light in the surroundings of the resonant nanoparticle [32]. Unfortunately, plasmonic nanosensors are mainly based on the spectral shift of the plasmon resonance as a function of external stimuli requiring a spectral analysis and then the use of tunable laser sources. Additionally, all-optical sensors based on plasmonic phenomena are affected by ohmic losses. For this reason, high-refractive index nanoparticles are a suitable alternative for this kind of task [33].

The aim of this work is to search for an alternative or an improvement over the existing sensing principles that simplifies the overall system by proposing a displacement sensor based on two parallel silicon nanowires with a section such that they are both resonant at a certain incident wavelength. While the excited resonances produce a remarkable increment of the scattered field, the distance and consequent interaction between the nanowires produce a detectable change in the spatial distribution of this field. Both the illumination and the readout systems are conceptualized as simple. In the case of illumination, the nanowires have been designed to be resonant at the wavelength of conventional infrared commercial lasers. On the other hand, the displacement is read by measuring the scattered field at that wavelength in only one direction, avoiding the incident beam.

The proposed design has been optimized in terms of the geometrical parameters for two different incident wavelengths in the infrared range and the measurement of displacement values compatible with vibration sensors in mechanical applications. The simulated results show high sensitivity, linearity, and reliability, making it suitable for realistic applications. Additionally, those distance ranges can be tuned by redesigning the section of the nanowires.

## 2. Materials and Methods

The considered structure and related simulation setup are constituted by two parallel suspended Si nanowires (see scheme in Figure 1a) with a diameter (*d*) of 140 nm and a length large enough to be considered infinite in our numerical simulations (*L* ≥ 5 × *λ*, *λ* being is the wavelength of the incident radiation). For this study, previous simulations were carried out with different values of *L* to properly establish a limit such that the response is insensitive to the length. The actual length (*L*) of the nanowires considered in the simulations is *L* = 5 × *λ* in order to minimize the time and simulation resources required by COMSOL Multiphysics. This long length also allows us to avoid the inclusion of the anchoring effects because we are considering the central part of long nanowires. This proposed geometry is affordable using nanofabrication techniques in state-of-the-art, such as electron beam (ELB) or focused ion beam (FIB) lithography [34]. The theoretical optical material parameters for silicon have been taken from Palik [35].

As mentioned before, the geometrical properties of the nanowires, as well as the incident wavelength, have been previously optimized to give rise to Mie resonances [36] in the near-infrared (NIR) spectral region to observe a strong variation of the scattered field by changing the separation distance between the two nanocylinders. The variation of the distance between the nanowires due to mechanical vibrations (*D*) is responsible for producing a variation in the resonant optical coupling between them (Figure 1b) producing a detectable change of the scattered field at the detection point.

The simulations were performed using a finite-element method implemented in the commercial software COMSOL Multiphysics. In particular, we used the Optics Waves Module that allows us to formulate and solve the differential form of Maxwell’s equations (in the frequency domain) together with boundary conditions. The two nanowires were placed at the center of a spherical homogeneous region filled with air, whose radius is 10 × λ (Figure 2). A perfectly matched layer (PML) domain, with a thickness of 2 × *λ* was placed outside the embedding medium domain. The PML layer acts as an absorber for the scattered field. In addition, an inner sphere of radius 6 × *λ* was added to place the electric field sensing points. This sensing sphere is located close to the nanostructures to avoid the occurrence of noise or other disturbances that may arise from placing it at greater distances. It is not recommended to sense on the surface of the PML since erroneous results tending to infinite values can be obtained. The mesh was chosen sufficiently fine to allow numerical convergence of the results. In particular, the element size of the mesh of the embedding medium was smaller than *λ*/5, and that of the nanowires was smaller than λ/[3ℜ(n)], *n* being the silicon refractive index.

The described system was illuminated with an incident plane wave polarized along the *x*-axis and propagating along the negative direction of the *z*-axis (red arrow in Figure 1a). Illumination with a plane wave propagating in the negative direction of the *z*-axis and linearly polarized to the *y*-axis (polarization perpendicular to the axis) is not presented here because simulated results show no clear and detectable change in the spatial distribution of the scattered field despite the variation in the distance between the nanowires as in the other polarization state.

The chosen illumination wavelength (*λ*) corresponds to 1064 nm or 1310 nm, emission wavelengths of commercial infrared (IR) lasers, which are cheaper than tunable lasers. The scattered field was detected in the central region of the air layer surrounding the nanowires at 90° to the illumination direction (green arrow in Figure 1a). From an experimental point of view, the scattered field is not easy to measure due to the difficulty of separating the incident from the scattered signal. Measuring the scattered signal at 90° is a good way of avoiding the signal coming from the incident field [37]. In addition, the sensing point has been fixed laterally to the axis joining the two cylinders (in the direction of the *y*-axis in Figure 1a). Both the illumination and detection systems would be performed with either lensed optical fibers or shaped optical fibers providing sub-micron spot sizes for a futurist prototype of the proposal [38].

Our main objective is to accurately reproduce the evolution of the scattered field in the far field of the nanosystem as a function of the separation distance (*D*), which varies due to the mechanical displacements. To obtain a strong confinement of the electromagnetic radiation (hot-spot) between the nanowires (*gap*), small distances are required [39,40,41]. Considering this fact and conventional vibrational distances, the smallest spacing considered in this work was 200 nm.

## 3. Results and Discussion

In order to attain the resonant behavior at 1064 nm or 1310 nm, the radius of the cylinder was fixed to 70 nm after a previous optimization process. A parametric sweep of the distance (*D*) from 200 nm to 600 nm with a step of 50 nm was performed.

To corroborate the suitability of the selected detection point and to obtain reliable far-field scattering results, a convergence test was performed, and the detection point on the inner auxiliary sphere was chosen. It is found that placing sensing points on the surface of the PML can lead to numerical errors, especially when a single point is involved.

### 3.1. Case Study for Operation Wavelength at 1064 nm

The first case of study corresponds to the wavelength operation at 1064 nm.

Any vibrational movement due to the excitation of a mechanical resonance will produce a periodic change in the *gap* distance and, consequently, a detectable change in the scattered field. The optical response (detected scattered field in far-field) as a function of the distance between the nanowires presents a global maximum and a global minimum such that it allows a qualitative analysis of the vibration. Figure 3a shows the modulus of the scattered field at the detection point as the function of the *gap* distance. This curve presents two minimum points at 450 nm and 1600 nm and a maximum point at 825 nm. After the second minimum at 1600 nm, the behavior is repeated, and the other maximum point is reached around 1900 nm. Therefore, we have focused on the analysis of results obtained in the distance range from 400 nm to 1600 nm. Regardless of their maximum o minimum points conditions, other points of interest have been selected to attend on the basis of the following issues:(1)It is not recommended to work with variations of the distance between the nanowires greater than 400 nm because the deformation of the nanocylinders can cause second harmonics.(2)The identification of intervals with linear behavior is highly desirable, and according to the first recommendation, it is not necessary to consider the whole range. In our case, selecting small displacement intervals with the best possible linear behavior is sufficient.

The near-field maps at the *gap* distances marked in Figure 3a are depicted in Figure 3b–e, where a change of the scattered field in the detection direction is observed.

The near-field pattern shown in Figure 3b (label 1 in Figure 3a) corresponds to the scattered field emitted by an electric dipole excited in the y-direction. For that reason, in the detection point, far-field along the *y*-axis, it is not possible to detect the signal. For labels 2 (Figure 3c) and 3 (Figure 3d), different patterns can be observed, increasing the amount of radiation that is scattered along the *y*-axis. Finally, in case four (Figure 3e), the distribution has a more complex shape, and at the same time, the scattered field distribution in the *y*-axis direction decreases. From 1600 nm, the previous pattern of a minimum followed by a maximum is found again.

In agreement with the above explanation and to facilitate the interpretation of the results, Table 1 shows the behavior of the scattered field in three different *gap* intervals.

Regarding the norm of the scattered field versus Si nanowires distance, we see the largest variations for interval III, but this is not the only aspect to consider. As a parameter of interest, we also analyze the sensitivity. For the design of an effective optomechanical sensor, high sensitivity values are necessary for accuracy during the measurement process. However, the higher the sensitivity, the narrower the measuring range, and the worse the stability. According to Equation (1), the sensitivity of a standard sensor is expressed as the change in the output value (in our work, detected scattered field) as a consequence of variations in the input value (displacement between the nanowires due to the mechanical movement).
(1)$$S=\frac{\Delta \left(\mathrm{Sensor}\;\mathrm{output}\;\mathrm{value}\right)}{\Delta \left(\mathrm{Sensor}\;\mathrm{input}\;\mathrm{value}\right)}$$

This suggests that for our particular study, Equation (1) can be expressed as Equation (2).
(2)$$S=\frac{\Delta E}{\Delta d}\times \left[V/m^{2}\right]$$

As can be seen in Table 2, the highest sensitivity was obtained for “Interval I.” It should be noted that high sensitivity values were achieved for the three different analytical regions (intervals I-III), keeping a similar order of magnitude among all of them.

Although large sensitivity values are necessary, a linear response of the output signal (scattered field) with respect to incident one (the *gap* between the nanowires) is required in the design of the sensor. To analyze the linearity of the proposed device, we have made a linear curve fitting for the three different intervals and have compared the coefficient of determination (*R*^2^). In all the cases, *R*^2^ takes values close to 1, confirming the linear response of the scattered field to changes in the *gap* between the nanowires.

### 3.2. Case Study for Operation Wavelength at 1310 nm

For the study with an operating wavelength of 1310 nm, the same procedure as in the previous case (Section 3.1) was followed. The dimensions of the nanowires were set as specified above, and a sweep of the nanocylinder distance (*D*) values was performed from *D* = 200 nm to *D* = 2200 nm with a step of 50 nm. Figure 4a shows the norm of the scattered field as a function of the displacement. Again, the behavior is the following, after reaching an absolute minimum for separation of 660 nm (pointed out in Figure 4a with label 1), it begins to increase almost constantly until reaching a maximum value at 950 nm (Figure 4c with label 2). Then, the behavior of reaching a minimum and increasing again is repeated.

In comparison with the previous one, different near-field patterns were obtained for this wavelength due to variations in the scattered field distribution. The first ones are similar to the case of *λ* = 1064 nm, but as the separation between the nanowires increases, a symmetry with respect to the *y*-axis is generated, as seen in Figure 4e.

As in the previous study, three well-differentiated intervals emerge in which good linearity with well-defined maxima and minima can be seen. These three intervals are summarized in Table 3.

In this case, slightly more pronounced maximum and minimum values are observed with respect to the previous study (*λ* = 1064 nm), leading to the sensitivity values of Table 4, according to Equation (2).

A linear fitting was carried out for each interval corresponding to the operation wavelengths of 1064 nm and 1310 nm, respectively. The coefficient of determination (*R*^2^) concludes that the linearity is acceptable, and the coefficients of determination are close to 1 in most cases. The approximations to the corresponding straight lines in each section can be seen in Figure 5. It is clearly observed that the numerical points fit well to a linear response in both cases.

## 4. Conclusions

These days, the trend is to design smaller, lighter, and integrable sensors with the simplest operation principle to reduce the power and space requirements. In this sense, all-optical nanosensors are one of the most promising alternatives to achieve this target.

This work demonstrates the potential application of light resonant nanowires to measure mechanical displacement by analyzing the light scattering signal. The proposed design is based on two parallel nanowires illuminated by a plane wave. These nanowires are composed of a high refractive index dielectric material, particularly silicon, providing a resonant signal in the visible or near-infrared region. In this case, the geometrical properties of the nanowires have been optimized in order to observe this resonant effect at the wavelength of a typical commercial infrared laser (1064 nm and 1310 nm). The geometrical properties of the nanowires are affordable to the current nanofabrication state of the art. The scattered field is numerically measured in a perpendicular direction to the illumination one to remove the incident signal. The resonant phenomenon and their coupling effects strongly depend on the distances between the nanowires, giving us information about this parameter.

The observed results show that the scattered electric field has a periodic variation as a function of the *gap* distance between the two nanowires. These variations are characterized by a maximum and a minimum signal with linear behavior between them at certain distance intervals. In this sense, we have obtained a maximum sensitivity of 1.14 × 10^6^ V/m^2^ considering a distance *gap* in the interval between 650 and 950 nm and under an illumination wavelength of 1310 nm. On the other hand, the best sensitivity at 1064 nm has a value of 1.1 × 10^6^ V/m^2^ in the interval between 500 and 750 nm, so both lasers are competitive. Additionally, in both cases of operating wavelengths and for most of the intervals presented in Figure 5, a coefficient of determination (*R*^2^) larger than 0.9 can be seen, demonstrating the linear fit of the model and its reliability. This behavior is reproducible at larger *gap* distances but with smaller sensitivities.

We can conclude that the proposed sensing principle provides interesting results about sensitivity and linearity in measuring mechanical displacement using an all-optical signal. In addition, the design operates with commercial infrared lasers, which are much cheaper than supercontinuum lasers. Both the illumination and detection system can be implemented using small spot size systems, such as lensed optical fibers, providing an illuminated spot on the center of the nanowires and avoiding anchoring and non-uniform effects. This system could have applications in vibration detection in mechanical engineering, predictive maintenance, or system monitoring, among others.

## Figures and Tables

**Figure 1 nanomaterials-12-04107-f001:**
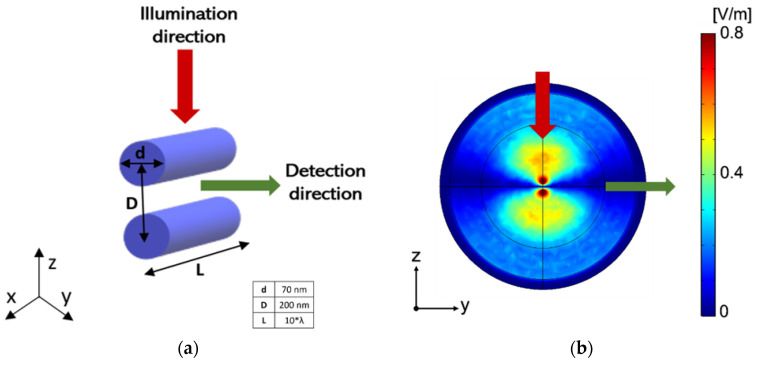
(**a**) Scheme of the proposed sensor based on two parallel silicon nanowires and its illumination configuration. (**b**) Near-field map of the norm of the scattered field in the YZ plane. The *gap* distance is 200 nm. Illumination with a plane wave propagating along the negative direction of the *z*-axis and linearly polarized parallel to the axis of both nanocylinders (*x*-axis).

**Figure 2 nanomaterials-12-04107-f002:**
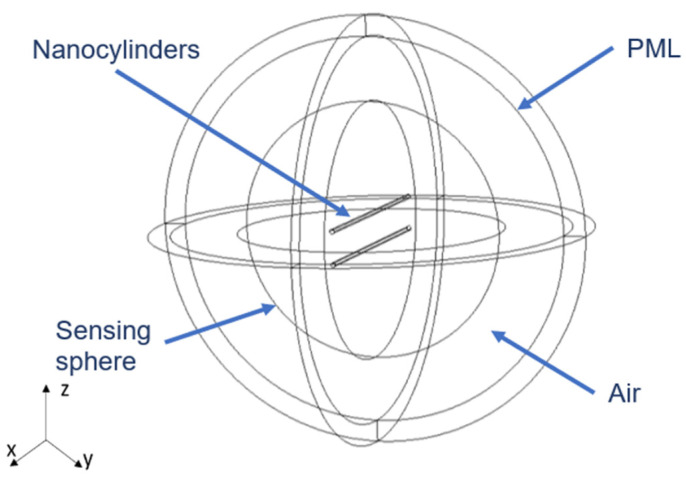
Scheme of the structure modeled in COMSOL.

**Figure 3 nanomaterials-12-04107-f003:**
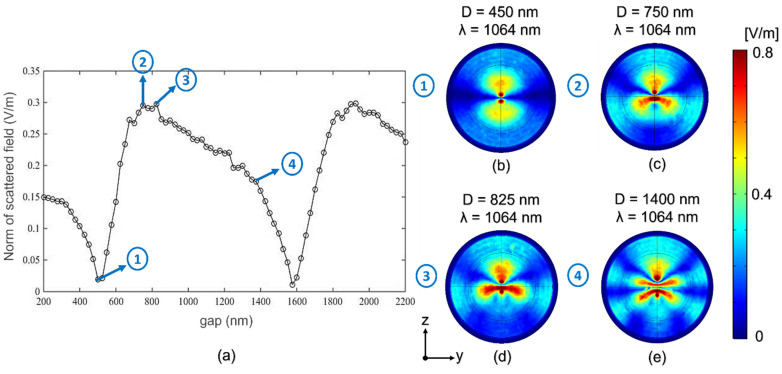
(**a**) Norm of the scattered field versus distance between nanocylinders for an incident wavelength *λ* = 1064 nm. Scattered field patterns at (**b**) *D* = 450 nm. (**c**) *D* = 750 nm. (**d**) *D* = 825 nm. (**e**) *D* = 1400 nm.

**Figure 4 nanomaterials-12-04107-f004:**
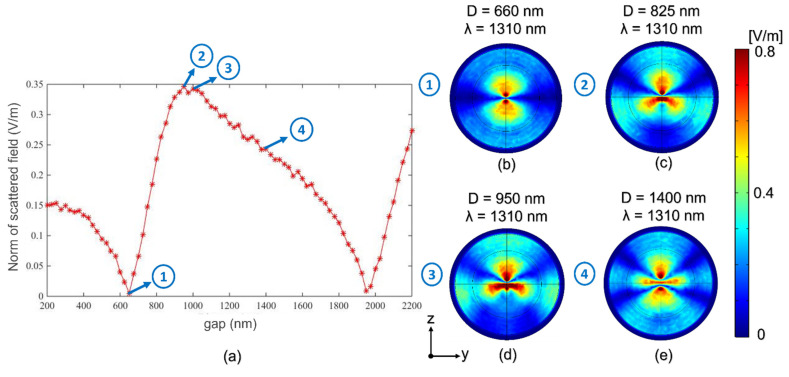
(**a**) Norm of the scattered field versus distance between nanocylinders for an incident wavelength *λ* = 1310 nm. Scattered field patterns at (**b**) *D* = 660 nm. (**c**) *D* = 825 nm. (**d**) *D* = 950 nm. (**e**) *D* = 1400 nm.

**Figure 5 nanomaterials-12-04107-f005:**
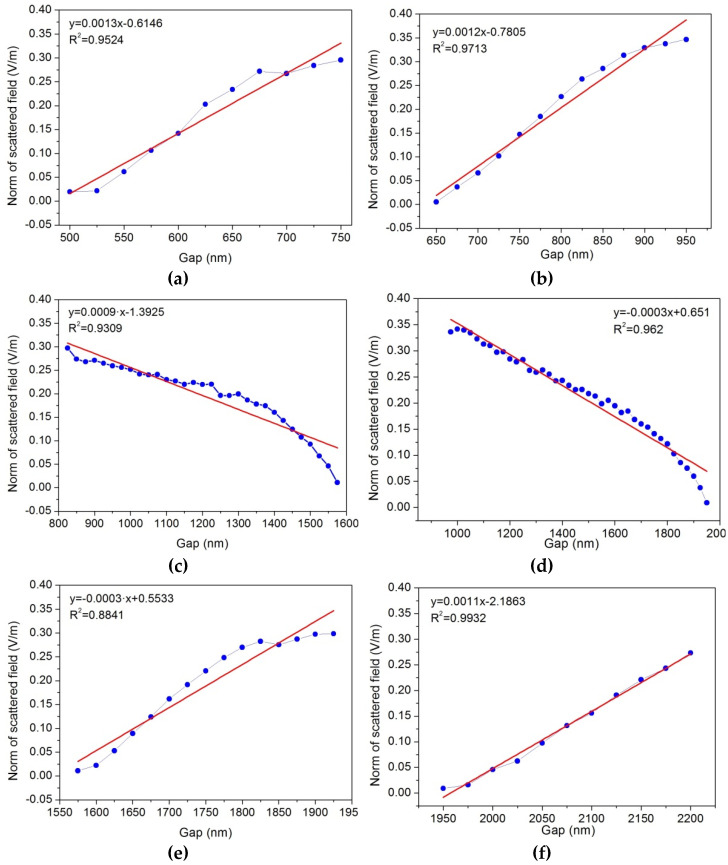
Linear fittings of the norm of the scattered field for 1064 nm. (**a**) Interval I. (**c**) Interval II. (**e**) Interval III. Linear fittings of the norm of the scattered field for 1310 nm. (**b**) Interval I. (**d**) Interval II. (**f**) Interval III.

**Table 1 nanomaterials-12-04107-t001:** Most significant intervals of the scattered field for *λ* = 1064 nm.

		Distance between Nanowires (nm)	Electric Field (V/m)
1064 nm	Interval I	500–750	0.019–0.295
	Interval II	825–1575	0.297–0.011
	Interval III	1575–1925	0.011–0.298

**Table 2 nanomaterials-12-04107-t002:** Sensor sensitivity for *λ* = 1064 nm.

		$\mathrm{Sensitivity}\;(V/m^{2})$
1064 nm	Interval I	$1.10\times 10^{6}$
	Interval II	$\;3.81\times 10^{5}$
	Interval III	$\;8.21\times 10^{5}$

**Table 3 nanomaterials-12-04107-t003:** Most significant intervals of the scattered field for *λ* = 1310 nm.

		Distance between Nanowires (nm)	Electric Field (V/m)
1310 nm	Interval I	650–950	0.005–0.346
	Interval II	975–1950	0.336–0.009
	Interval III	1950–2200	0.009–0.273

**Table 4 nanomaterials-12-04107-t004:** Sensor sensitivity for *λ* = 1310 nm.

		$\mathrm{Sensitivity}\;(V/m^{2})$
1310 nm	Interval I	$1.14\times 10^{6}$
	Interval II	$3.36\times 10^{5}$
	Interval III	$1.06\times 10^{6}$

## Data Availability

Not applicable.

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
