# Peer review of "All-Optical Nanosensor for Displacement Detection in Mechanical Applications"

_nanomaterials, 2022, doi:10.3390/nano12224107_

Round 1
Reviewer 1 Report
In this paper the authors propose the design of an optical system based on two parallel resonant Silicon nanowires as base of an all-optical displacement sensor. The variation of the distance between the nanowires produces a detectable change in their resonant behavior and optical interaction. The proposed design aims to use a simple light source by using a commercial diode laser and to simplify the readout systems whit high linearity and sensitivity.
The manuscript is well written and easy to be read, anyway, the technical level and the level of novelty is rather low at this stage
In particular the model is based on 2 wires in air where it is not clear where they are anchored or how this can be fabricated.
moreover, and more important, it is really not clear how this device should work, i.e. how in a real experiement the displacement can be detected. The authors report here only the scattering field, but a more clear and measurable unit should be used (for example transmittance etc.)
At this stage the manuscript cannot be accepted for publication, but with proper improvements it can be reconsider. The idea must be more close to a "real" device and application
I also recommend to improve the introduction mentioning for example all the class of nanodevices based on metallic nanostructures able to detect mechanical effect (a good review on that is Adv. Funct. Mater. 2021, 2103706, DOI: 10.1002/adfm.202103706)
Author Response
"Please see the attachment."

Reviewer 2 Report
The authors present a theoretical study on a new design of optical nanosensor based on two parallel resonant silicon nanowires which could have applicability in studying mechanical displacements directly correlated with the changes induced in the observed electric field. The subject is interesting and appealing however the manuscript needs to be improved before being published. The following issues must be addressed:
1. The abstract needs to be clearer and present more details about the work as well as the novelty elements.
2. Which is actually the length (L) of the wires considered in the simulations? How does a different value for the diameter if the wire (d) influences the sensitivity? It would be interesting to study one more dimension of nanowires.
3. How does the direction of the polarization - perpendicular to the axis or linear polarization- influence the results?
4. The obtained sensitivity should be compared to previous values from literature for similar optical sensors.
5. he authors mention several times the linearity of the readout however there is no plot to confirm this, the reader can only guess it form the values.
6. In Table 2 the values for sensitivity for intervals II and III are reversed. Please correct.
7. The maximum sensitivity values obtained for the two lasers are not for the same interval, it is 250 nm for 1064 and 300 nm for 1310 nm. Please correct this.
8. The manuscript needs to carefully checked as there are some typing errors present.
Author Response
"Please see the attachment."

Reviewer 3 Report
The authors demonstrate how the scattering field depends on the distance between Si nanowires. As this dependence is strong at the resonance frequency, they propose to use the scattering light to detect a displacement of the wires. This manuscript is well written and technically well done. However, the scientific soundness and interest to readers seems low. In order to stress the significance of the obtained results, some clarifications and discussions should be added in accordance with the following comments:
1) Please, give the references to the experimental papers where the changing in the distance between two nanowires should be measured, and explain why it is so important to do such kind of measurements.
2) Please, explain why the Si-nanowires were taken. Please give a reference of experiments where similar system of two Si-nanowires of length of 5 um and diameter of 70 nm is used.
3) This task seems to be more engineering than scientific. Such kind of research is more appropriate to do for specific conditions of a specific experiment.
There are some misprints.
1) In abstract “whit high linearity and sensitivity”.
2) In Fig. 1, “Illumination with a plane wave propagating along the negative direction of the z-axis and linearly polarized along the axis that joins both nanocylinders (x-axis).” Probably, should be “linearly polarized perpendicular to the axis that joins both nanocylinders”
3) Same information appears twice in “The chosen illumination wavelength (λ) is 1064 nm or 1310 nm, which correspond to the emission wavelengths of commercial lasers” and in “The operating wavelengths corresponded to 1064 nm and 1310 nm, corresponding to infrared (IR) lasers of commercial values on the market and cheaper than white light lasers”.
Author Response
"Please see the attachment."

Round 2
Reviewer 1 Report
I think that the manuscript is now almost ready to be accepted for publication. anyway, the quality of the figures are really low and MUST be improved, in particular the plots that are hard to be read
Author Response
"Please see the attachment."

Reviewer 2 Report
The authors have addressed all the comments raised and included the missing data in the manuscript. The manuscript can be accepted for publication after improving the quality of the new figure 6 inserted and replacing in the conclusion section the statement "remarkable linearity" because it is not quite the case here as seen in Figure 6.
Author Response
"Please see the attachment."

Reviewer 3 Report
From authors’ response to my first report, I conclude that there are no experimental papers where the distance between two long-length suspended nanowires should be measured in the narrow range below 1um. The idea to measure interparticle distance using plasmon resonance is not novel and already used in [26]. The calculation done by authors would be great as a part of an experimental work with specific design taking into account all the details of the experimental setup. I think the main proposal of the authors is to adopt the diameter of nanowire for plasmonic resonance at specific laser wavelength and to do measurement at specific wavelength. This idea seems obvious. The calculation of the scattering field versus the interparticle distance and further optimization are a pure engineering task. To my mind, this paper does not fit for high level scientific journal.
Author Response
"Please see the attachment."
